# SPECIAL SOLUTIONS WITH SMALL VOLUME EXIST

**Tausifa Jan Saleem,**[*] **Ramanjit Ahuja,**[†] **Surendra Prasad, Brejesh Lall**
Bharti School of Telecommunication Technology and Management
Indian Institute of Technology Delhi
Delhi, India

## ABSTRACT

The Lottery Ticket Hypothesis Frankle & Carbin (2018) for deep neural networks emphasizes the importance of initialization used to re-train the sparser networks obtained using the iterative magnitude pruning process. An explanation for why the specific initialization proposed by the lottery ticket hypothesis tends to work better in terms of generalization (and training) performance has been lacking. In this work, we attempt to provide insight into this phenomenon by empirically studying the volume/geometry and loss landscape characteristics of the solutions obtained at various stages of the iterative magnitude pruning process.

## 1 INTRODUCTION

Neural network pruning is the process of removing unnecessary weights from a neural network Blalock et al. (2020). Pruning reduces model size and energy consumption, making inference more efficient. However, it has been observed that pruned models do not perform well without re-training, and that re-training sparse networks from scratch using random initialization is difficult Han et al. (2015); Li et al. (2016). Frankle & Carbin (2018) demonstrated that there exists a subnetwork which, when trained from the start with a specific initialization, can reach the accuracy of the original dense network. More formally, their hypothesis states that a randomly initialized dense neural network contains a subnetwork that is initialized such that when trained from scratch, it can match the test accuracy of the original network after training for at most the same number of iterations. This hypothesis has been named as *lottery ticket hypothesis* and has been empirically validated using a procedure known as *Iterative Magnitude Pruning (IMP)* Frankle & Carbin (2018); Frankle et al. (2019). IMP has been successful in producing highly sparse networks that retain the performance of their dense counterparts. Despite this success, the underlying principles governing why IMP works remain poorly understood. Paul et al. (2022) studied the geometry of the error (test error) landscape to better understand the lottery ticket hypothesis and IMP. In this work, we instead study the characteristics of the *loss* landscape along with the volume/geometry of solutions obtained at different stages of the IMP procedure. While training loss and test error are correlated, analyzing the loss landscape is more directly relevant for understanding optimization dynamics, since stochastic gradient descent (SGD) navigates the loss landscape rather than the error landscape.

**Contributions.** We demonstrate the existence of a special type of solutions in the loss landscape that generalize well yet occupy an extremely small volume in the original parameter space. The IMP procedure exposes these solutions, which would otherwise remain hidden. This observation provides a new insight into the lottery ticket hypothesis and IMP, helping to explain how pruned networks can retain high performance despite significant sparsity. Given that IMP is widely used in edge AI applications, understanding its efficiency and theoretical and/or conceptual underpinnings is critical for optimizing deployment strategies.

---

[*]Corresponding author (tausifa.cstaff@iitd.ac.in). This work was accomplished during the author's affiliation with IIT Delhi. She is currently affiliated with MBZUAI as a Postdoctoral Researcher.

[†]This work was accomplished during the author's affiliation with IIT Delhi. He is currently affiliated with ON Semiconductor Corp (onsemi) as Member of Technical Staff.

## 2    DEFINITIONS AND NOTATIONS.

**Sparse subnetworks:** Given a dense network with weights $W$ ($W \in \mathbb{R}^D$), a sparse subnetwork has weights $m \odot W$, where $m \in \{0,1\}^D$ is a binary mask and $\odot$ is the element-wise product. The sparsity of a mask $m$, $S(m)$ is the fraction of zeros in the mask.

**Notation for IMP solution at level** $L$**:** We represent the IMP solution (minimum) at level $L$ by $W_{(L)}^{(min\_(L))}$, weights of the dense network at initialization by $W^{(init)}$ and weights of the dense network at rewind-point by $W^{(rewind\_point)}$. Note that all these weights are $D$-dimensional.

**Projection of level** $L$ **solution on level** $(L+1)$**:** Let $W_{(L+1)}^{Pr(min\_(L))}$ represent the projection of level $L$ solution on level $(L+1)$. It is obtained as $W_{(L+1)}^{Pr(min\_(L))}=m_{(L+1)} \odot W_{(L)}^{(min\_(L))}$, where $m_{(L+1)}$ represents the pruning mask at level $(L+1)$. For example, the projection of level 0 solution on level 1 will be represented as $W_{(1)}^{Pr(min\_(0))}$. Note that $S(m_{(L+1)}) > S(m_{(L)})$.

**Reverse Projection of level** $(L+1)$ **solution on level** $(L)$**:** Let $W_{(L)}^{RPr(min\_(L+1))}$ represent the reverse projection of level $(L+1)$ solution on level $(L)$. It is obtained as; $W_{(L)}^{RPr(min\_(L+1))}=m_{(L)} \odot W_{(L+1)}^{(min\_(L+1))}$, where $m_{(L)}$ represents the pruning mask at level $(L)$. For example, the reverse projection of level 1 solution on level 0 will be represented as $W_{(0)}^{RPr(min\_(1))}$.

## 3    RESULTS AND DISCUSSION

This section details a key experimental finding and its underlying rationale. We perform experiments on a widely used network, ResNet-20, using the benchmark dataset CIFAR-10. To assess the generality of our observations, we also performed experiments on VGG-16 trained on CIFAR-10 and observed consistent results. For brevity, the results on VGG-16, along with experimental details for both architectures, are provided in the Appendix. A plot of training loss and test accuracy at different levels of IMP for ResNet-20 is presented in Fig. 1.

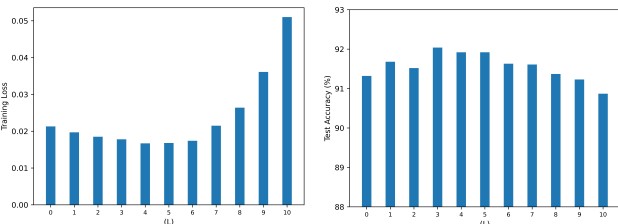

Figure 1: Training loss and test accuracy at different levels of IMP. **Left:** Training loss. **Right:** Test accuracy.

Huang et al. (2020) proposed that there exist two types of minima in the neural network loss landscape: *good minima* and *bad minima*. Good minima are characterized by low training loss, large basin volumes, and strong generalization performance. Bad minima also achieve low training loss but are associated with small basin volumes and poor generalization performance. We demonstrate that there also exist other kind of solutions [1] which have good generalization performance but have a (relatively) small volume. This implies that volume is not the only criterion for generalization performance; there is more to it. A careful experimental study leads us to hypothesize that the small volume of these solutions is due to a very sharp curvature in certain dimensions, but the coefficients in these dimensions are zero. Their *volume* measure tends to increase when these inferior dimensions are removed (possibly via pruning at another point), but is small when considered in the original space. **This makes these solutions undiscoverable by SGD in the original space but can be easily discovered in the pruned space.**

---

[1]We cannot say for sure that these solutions are true minima, but they lie in the neighborhood of minima because the gradient of the vast majority of parameters at these points is zero.

Consider two IMP solutions $W_{(L-1)}^{(min\_(L-1))}$ and $W_{(L)}^{(min\_(L))}$ at levels $(L-1)$ and $L$, respectively. $W_{(L-1)}^{(min\_(L-1))}$ is a baseline for $W_{(L)}^{(min\_(L))}$ because the pruning mask for level $L$ is determined by $W_{(L-1)}^{(min\_(L-1))}$. A comparison of the euclidean distance between $W_{(L)}^{Pr(min\_(L-1))}$ and $W_{(L)}^{Pr(rewind\_point)}$, and the euclidean distance between $W_{(L)}^{(min\_(L))}$ and $W_{(L)}^{Pr(rewind\_point)}$ given in Figure 2 shows that for all $L$ except 2 and 3, $W_{(L)}^{Pr(min\_(L-1))}$ is closer to $W_{(L)}^{Pr(rewind\_point)}$ than $W_{(L)}^{(min\_(L))}$. Despite this, at level $(L)$ SGD converges to $W_{(L)}^{(min\_(L))}$ and not to $W_{(L)}^{Pr(min\_(L-1))}$.

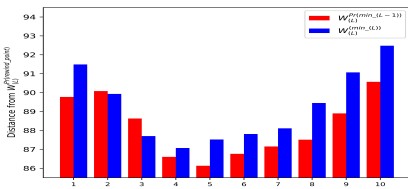

Figure 2: Distance from $W_{(L)}^{Pr(rewind\_point)}$ to $W_{(L)}^{Pr(min\_(L-1))}$ and to $W_{(L)}^{(min\_(L))}$.

To find the underlying reason, we plot the SGD trajectory for level $L$ and that for level $(L-1)$ projected on level $L$, which is shown in Figure 3. It is evident from the figure that the trajectory

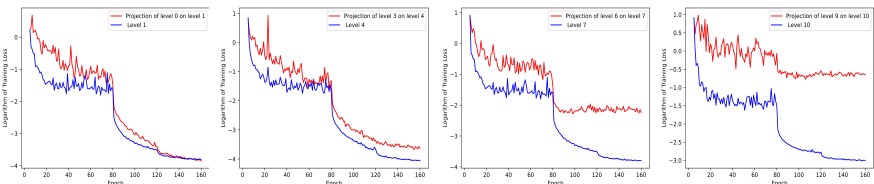

Figure 3: Comparison of logarithm of training loss versus epoch between level $(L)$ and level $(L-1)$ projected on level $(L)$.

for level $(L)$ is steeper than for level $(L-1)$ projected on level $L$. This makes SGD converge to $W_{(L)}^{(min\_(L))}$ and not to $W_{(L)}^{Pr(min\_(L-1))}$. To approximate the volume of a basin around a minimum, we follow the approach of Wu et al. (2017), which estimates the inverse basin volume using the top-$k$ positive eigenvalues of the Hessian of the loss function. Specifically, the inverse volume $V'(k)$ is approximated as $V'(k) := \sum_{i=1}^{k} \log(\lambda_i)$, where $\{\lambda_i\}_{i=1}^{k}$ denote the top-$k$ positive eigenvalues of the Hessian. This approximation is motivated by the observation that larger basins correspond to flatter minima, which are characterized by smaller Hessian eigenvalues, and vice-versa.

Using this approximation with $k = 100$, we compare the basins surrounding $W_{(L)}^{\min(L)}$ and $W_{(L)}^{Pr(\min(L-1))}$. We observe that the basin around $W_{(L)}^{\min(L)}$ has a larger volume than that around $W_{(L)}^{Pr(\min(L-1))}$, as shown in Figures 4 and 5.

Next, if we consider the reverse projection of level $(L)$ solution on level $(L-1)$, $W_{(L-1)}^{RPr(min\_(L))}$ and the level $(L-1)$ solution, $W_{(L-1)}^{(min\_(L-1))}$, the volume of basin around $W_{(L-1)}^{(min\_(L-1))}$ is seen to be larger than the volume of basin around $W_{(L-1)}^{RPr(min\_(L))}$. This is demonstrated in Figure 6 and Figure 7. And this explains why SGD does not converge to $W_{(L-1)}^{RPr(min\_(L))}$ at level $(L-1)$. To sum up the discussion above, SGD converges to $W_{(L)}^{(min\_(L))}$ at level $(L)$ instead of $W_{(L)}^{Pr(min\_(L-1))}$ because the volume of the basin around $W_{(L)}^{(min\_(L))}$ is much larger than the volume of the basin around $W_{(L)}^{Pr(min\_(L-1))}$, and the path to $W_{(L)}^{(min\_(L))}$ is steeper than the path to $W_{(L)}^{Pr(min\_(L-1))}$.

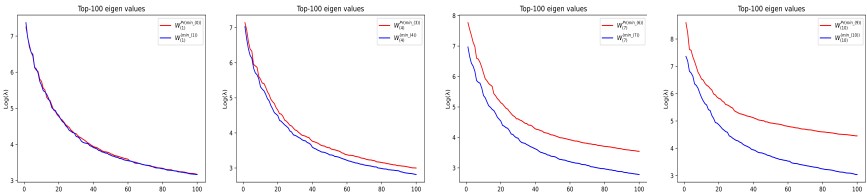

Figure 4: Comparison of top-100 positive eigenvalues of the Hessian at $W_{(L)}^{(min\_(L))}$ and $W_{(L)}^{Pr(min\_(L-1))}$. The figure shows that the eigenvalues of the Hessian at $W_{(L)}^{(min\_(L))}$ are smaller than those at $W_{(L)}^{Pr(min\_(L-1))}$.

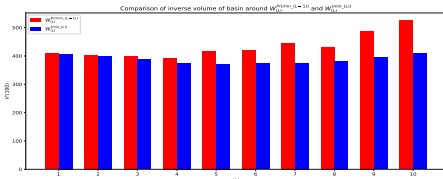

Figure 5: Comparison of inverse volume of basin, $V'(100)$ at $W_{(L)}^{Pr(min\_(L-1))}$ and $W_{(L)}^{(min\_(L))}$.

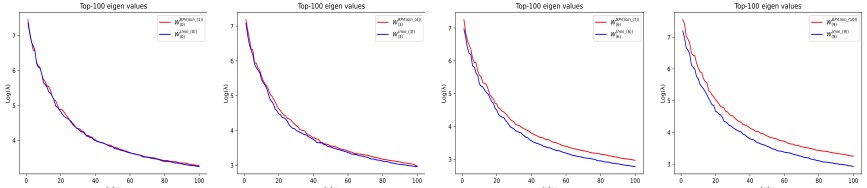

Figure 6: Comparison of top-100 positive eigenvalues of the Hessian at $W_{(L-1)}^{(min\_(L-1))}$ and $W_{(L-1)}^{RPr(min\_(L))}$. The figure shows that the eigenvalues of the Hessian at $W_{(L-1)}^{(min\_(L-1))}$ are smaller than that at $W_{(L-1)}^{RPr(min\_(L))}$.

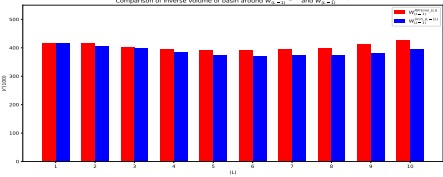

Figure 7: Comparison of $V'(100)$ at $W_{(L-1)}^{RPr(min\_(L))}$ and $W_{(L-1)}^{(min\_(L-1))}$.

However, in the $(L-1)$ space, the volume of the basin around $W_{(L-1)}^{RPr(min\_(L))}$ is much smaller than the volume of basin around $W_{(L-1)}^{(min\_(L-1))}$. **This means that the volume comparison gets flipped in the two spaces, $(L-1)$ space and $(L)$ space**.

## 4 CONCLUSION AND SCOPE FOR FURTHER WORK

In this work, we have studied the loss landscape characteristics and volume/geometry of the IMP solutions at different levels in order to provide insight about the lottery ticket hypothesis and IMP.

The study provided the following important result: there exist special type of solutions in the loss landscape, which perform well but have a very small volume in the original space, and the IMP procedure uncovers such solutions. The future work would be to come up with an approach that uncovers the sparse-dimensioned special solutions directly without going through the computationally and time-intensive IMP process.

## ACKNOWLEDGEMENT

This work was supported by funding from IIT Delhi, Cadence Design Systems, and the Indian National Science Academy (INSA).

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

# A  BACKGROUND INFORMATION

This section discusses the following preliminaries: lottery ticket hypothesis, loss landscape of a neural network, and the role of volume in generalization performance.

**Lottery ticket hypothesis.** Lottery ticket hypothesis postulates that neural networks contain subnetworks that when trained from scratch reach the accuracy of the original network in a commensurate number of epochs. This hypothesis was proposed by Frankle & Carbin (2018). Their assertion is supported by the observation that IMP consistently discovers such subnetworks on small vision tasks by rewinding the weights of the subnetwork to the $0^{th}$ iteration of the original dense network. However, IMP fails on deeper networks. In follow-up work, Frankle et al. (2019) demonstrated that in the case of deeper networks, such subnetworks could be obtained by rewinding to $k^{th}$ iteration instead of rewinding to $0^{th}$ iteration after pruning, for a suitably chosen value of $k$.

**Loss landscape of a neural network.** Neural networks are trained using feature vectors $x_i$ and their corresponding labels $y_i$. In the training procedure, loss function $Loss(W)$ is minimized:

$$Loss(W) = \frac{1}{n}\Sigma_{i=1}^{n}l(x_i, y_i, W) \tag{1}$$

where $W$ represents the network parameters, $n$ represents the number of input data samples, and $l(x_i, y_i, W)$ is a function that measures the difference between the predicted label and the actual label. The number of parameters in neural networks is very large; hence, the neural network loss functions reside in extremely high-dimensional spaces. The plot of training loss ($Loss(W)$) with respect to the network parameters ($W$) is referred to as the loss landscape.

Studying the loss landscape of neural networks is important for understanding their behavior. The loss landscape of under-parameterized models has multiple isolated local minima Liu et al. (2020). The set of solutions of over-parameterized models, on the other hand, is generically a manifold of dimension $m - rn$ Cooper (2018), where $m$ is the number of parameters, $n$ is the number of input data samples, and $r$ is the number of output classes. This means that the density of minima in the loss landscape of over-parameterized models is very high, and the optimizer converges to one of them irrespective of where it starts at.

**Role of volume[2] in generalization performance.** Flatness is an indicator of network performance sensitivity to parameter perturbations. The minimum is flat if small changes to the parameters do not cause misclassifications. On the contrary, the minimum is sharp if small changes to the parameters cause a number of misclassifications, thereby increasing the value of the loss function. A number of studies have focused on establishing the relationship between the flatness/sharpness of minima and their generalization ability. The following presents a summary of those studies:

Hochreiter & Schmidhuber (1997) explained the relationship between the flatness of minima and their generalization ability using the Minimum Description Length (MDL) theory. They defined a flat minimum as a region in the weight space where the error remains approximately constant. Such a region requires less information for representation because of its lower complexity than a region where the error changes drastically (sharp minimum). According to MDL theory, lower complexity models have higher generalization ability. Chaudhari et al. (2019) have shown that the local minima discovered by the optimizers have a flat geometry for a range of deep neural network architectures irrespective of their structures, training strategies, and the input data. These flat regions are robust to perturbations (both data perturbations as well as parameter perturbations) and noise in the activations, which makes them generalize well. Keskar et al. (2016) have demonstrated that the large-batch methods are attracted towards sharp minima. They have shown that these minima have large positive eigenvalues of the Hessian (Hessian of the loss function) and do not have good generalization ability. In contrast, small-batch methods converge to flat minima, which have a large number of small eigenvalues of the Hessian and generalize well. Dinh et al. (2017) argued that the notions of flatness cannot be directly related to the generalization performance without taking certain precautions. Their argument is based on the following grounds: the loss function of a neural network with weights much larger than one may seem to be flat because parameter perturbations by one unit will have a very small consequence on the network performance. On the contrary, in a neural network with smaller weights than one, the same perturbation will drastically affect the network performance, making the loss function appear sharp. Knowing that neural nets are scale-invariant, the large-parameter network and the small-parameter network are the same in that the large-parameter

---

[2]For our discussion here, volume refers to the volume of minimum, not the volume of loss sublevel set.

network is a rescaled version of the small-parameter network. Thus, any discrepancies in the loss function plots are simply the result of the difference in the scales of the networks. Hence, it is crucial to apply perturbations in accordance with the scale of network parameters to have a correct notion of the flatness/sharpness of minima Li et al. (2018).

Huang et al. (2020) demonstrated that two types of minima exist in a neural network loss landscape. These are referred to as the so-called good minima and bad minima. Good minima exhibit a small training loss and a small generalization error. Bad minima, too have a small training loss but exhibit a high generalization error. They also studied the qualitative difference in the loss landscape around these minima and observed that the decision boundaries of good minima have wide margins[3] while the decision boundaries of bad minima have very narrow margins. Huang et al. (2020) also illustrated that the good minima reside in wide basins[4] that exhibit a large volume in the parameter space, while the bad minima reside in narrow basins that exhibit a much smaller volume. A larger volume also implies a higher probability of hitting the minima by SGD. Volumes of the minima are, therefore, good indicators of their robustness and can provide useful insights Huang et al. (2020).

## B  FURTHER EXPERIMENTAL DETAILS

Some finer details of the experiments conducted in this study are mentioned here for completeness.
**ResNet-20 on CIFAR-10.** We train ResNet-20 on CIFAR-10 for 160 epochs with SGD and a batchsize of 128. We use learning rate = 0.1, momentum = 0.9, and weight decay = 0.0001. The learning rate is decayed by a factor of 10 at 80 and 120 epochs. We use iterative magnitude pruning with weight rewinding (IMP-WR) and run 10 iterations of IMP. We prune 20% of the smallest magnitude weights in each iteration. The prunable parameters are the weights of the convolutional layers and the fully-connected layers.
**VGG-16 on CIFAR-10.** We train VGG-16 on CIFAR-10 for 160 epochs with SGD and a batchsize of 128. We use learning rate = 0.1, momentum = 0.9, and weight decay = 0.0001. The learning rate is decayed by a factor of 10 at 80 and 120 epochs. We use IMP-WR and run 12 iterations of IMP. We prune 40% of the smallest magnitude weights at each round. The prunable parameters are the weights of the convolutional layers and the fully-connected layers.

## C  RESULTS ON VGG-16/CIFAR-10

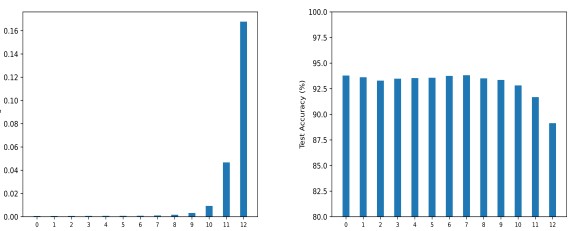

Figure 8: Training loss and test accuracy at different levels of IMP in case of VGG-16. **Left:** Training loss. **Right:** Test accuracy.

---

[3]Distance between the class boundary and the data.
[4]Set of points in the neighborhood of a minimum whose loss value is smaller than some cutoff value.

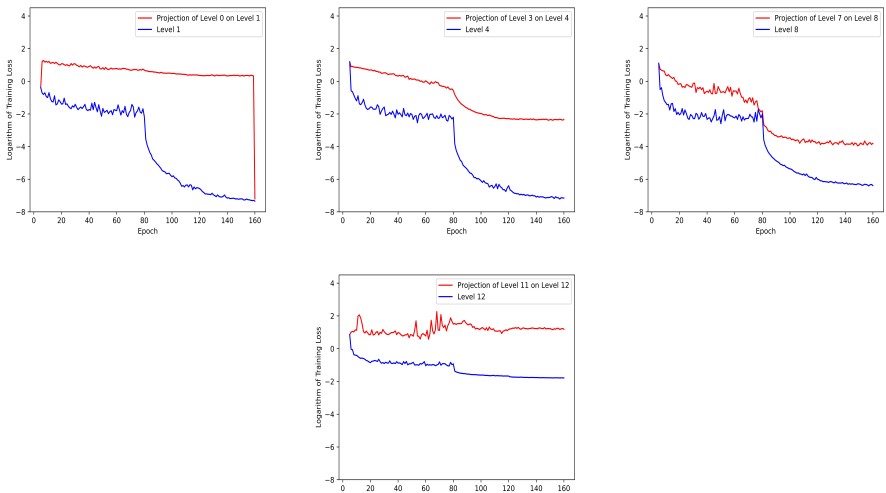

Figure 9: Comparison of the logarithm of training loss versus epoch between level ($L$) and level ($L-1$) projected on level ($L$).

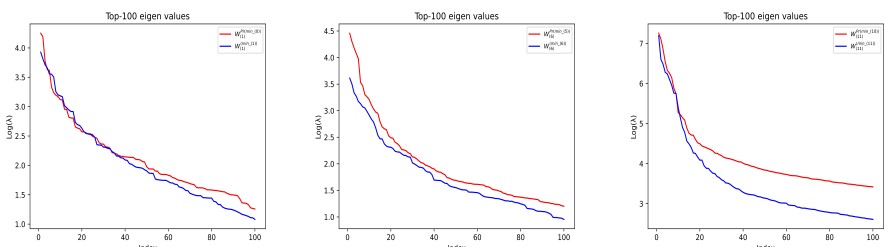

Figure 10: Comparison of top-100 positive eigenvalues of the Hessian at $W_{(L)}^{(min\_(L))}$ and $W_{(L)}^{Pr(min\_(L-1))}$.

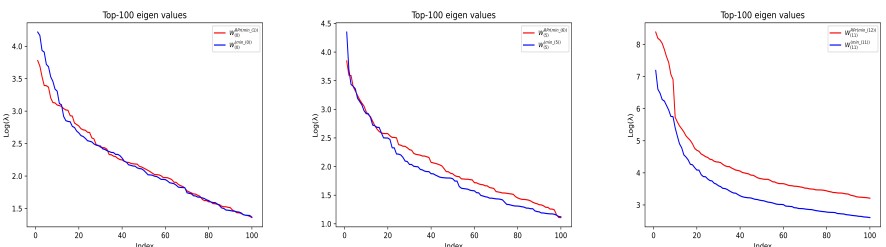

Figure 11: Comparison of top-100 positive eigenvalues of the Hessian at $W_{(L-1)}^{(min\_(L-1))}$ and $W_{(L-1)}^{RPr(min\_(L))}$.

