# OpenReview forum: "Special solutions with small volume exist"
_ICLR.cc/2026/Workshop/Sci4DL — Sci4DL 2026_

### Official Review · Reviewer_UcEH · 2026-02-09

**Fit:** 3
**Significance:** 2
**Confidence:** 2

**Summary:**

In this paper, the authors provided new insights into the lottery ticket hypothesis (LTH) and iterative magnitude pruning (IMP) via the loss landscape characteristics and volume/geometry of the IMP solutions. The authors show that IMP can uncover special solutions that generalize well while occupying a relatively small local volume in the original dense parameter space. These findings are demonstrated by experiments using ResNet20 and VGG16 trained on CIFAR10 dataset.

**Strengths:**

This paper is clear and easy to follow overall. And the authors provided a novel perspective by using IMP as a means to discover geometrically rare but with high performance solutions. The analysis on SGD trajectories, cross level projections, and basin volume derived from Hessian eigenvalue provide meaningful comparison of loss landscape characteristics across spaces. And that the results from two architectures consistently and jointly support the authors’ central empirical claim.

**Suggestions:**

Since the experiments shown in the paper are only in a rather small and clean CIFAR10 dataset, it would be interesting to see if such empirical findings can be obtained in larger or noisy datasets. Another potential future direction could be evaluating the proposed hypothesis on a different neural network architecture such as transformers which could help clarify the generality of the observed findings.

---

### Official Review · Reviewer_rvqD · 2026-02-26

**Fit:** 2
**Significance:** 2
**Confidence:** 2

**Summary:**

Iterative Magnitude Pruning (IMP) uncovers a special class of solutions in the loss landscape that generalize well but occupy an extremely small volume in the original parameter space. These solutions are essentially hidden from standard stochastic gradient descent (SGD) because their geometry makes them difficult to reach without pruning. The paper argues that IMP exposes these sparse-dimensioned solutions, which explains why pruned networks can retain high performance despite significant sparsity.

**Strengths:**

1. Puts forward an interesting hypothesis that a new class of solutions in the loss landscape can generalize well despite occupying very small volumes.
2.  Use of ResNet-20 on CIFAR-10 as the main experiment, supplemented by VGG-16 results in the appendix, shows thoughtful validation across architectures.
3. The paper show that the relative basin volumes of solutions can flip depending on whether they are considered in the original parameter space or the pruned space which is intuitive.
4. By analyzing Hessian eigenvalues and basin volumes, the paper provides a rigorous explanation for why SGD converges to certain solutions under IMP.

**Suggestions:**

1. Extending experiments beyond CIFAR-10 (e.g., ImageNet or NLP tasks) would strengthen claims of generality.
2. The claim "these solutions are undiscoverable by SGD in the original space but can be easily discovered in the pruned space" requires broader validation as there is a clearer distinction between “undiscoverable” vs. “hard to discover.” Other factors (like implicit regularization in SGD, learning rate schedules, or noise in optimization) should also be investigated.
3. Adding statistical measures of variability (across multiple seeds) would strengthen the empirical claims.
4. Comparisons with other works (if any) who worked on loss landscape characteristics and basin volume/geometry under IMP would strengthen the contribution.

---

### Official Review · Reviewer_13GC · 2026-02-27

**Fit:** 3
**Significance:** 2
**Confidence:** 2

**Summary:**

The paper analyzes the loss landscape volume of solutions found at different IMP iterations. Interestingly, solutions at level L and L+1 are compared in each other's spaces (e.g. solution L+1 in the space of L and vice-versa).
The primary finding is that there are good solutions (e.g. at Level L+1), which have a small volume in the original/previous space (e.g. at Level L). More specifically, the authors find that a well generalizing solution at level L has a larger volume than the solution at Level L-1 in the space of L. But the opposite is observed when comparing the volumes in the space of L-1.

**Strengths:**

The results are very interesting and shed some light on why IMP works well. I believe that this could lead to finding better pruning algorithms.
The experimental methodology appears sound.

**Suggestions:**

The notation is a bit unclear and the paper is difficult to follow in general.
I'd suggest the following:
* The definitions of the projection and reverse projection operators are defined in terms of L and L+1. However, in the results sections these are used with the notation of L and L-1 which adds confusion.
* It is unclear how the results in Figure 3 are generated. For e.g. the "projection of Level 3 on Level 4" curve in red, is the Level 3 solution at Epoch i projected to Level 4 and the loss then measured? Please clarify.
* The figures are very small and interfere with the flow of the text. (They also make the PDF lag on my machine). I'd suggest making these larger and moving some of them to the appendix.
* You compare "inverse volume" in the plots but then compare "volume" in the text, please be consistent.

---

### Meta-Review · Area_Chair_6nuE · 2026-02-28

**Recommendation:** Accept

**Metareview:**

This paper studies the properties of the loss landscape along pruning procedures in the context of the lottery ticket hypothesis. All reviewers noted the novelty and quality of the paper's contributions. I recommend acceptance.

---

### Decision · Program_Chairs · 2026-03-02

Accept